# Comparison at the first prenatal visit of the maternal dietary intakes of smokers with non-smokers in a large maternity hospital: a cross-sectional study

Eimer G O'Malley,[1] Shona Cawley,[1,2] Ciara M E Reynolds,[1] Rachel A K Kennedy,[1,2] Anne Molloy,[3] Michael J Turner[1]

[1]UCD Centre for Human Reproduction, Coombe Women and Infants University Hospital, Dublin, Ireland
[2]School of Biological Sciences, Dublin Institute of Technology, Dublin, Ireland
[3]School of Medicine, Trinity College Dublin, Dublin, Ireland

**Correspondence to**
Dr Eimer G O'Malley;
eimer.om@gmail.com

## ABSTRACT

**Objectives** Using detailed dietary and supplement questionnaires in early pregnancy, we compared the dietary intakes of micronutrients and macronutrients at the first prenatal visit of women who reported continuing to smoke during pregnancy with the intakes of women who were non-smokers.

**Design** Cross-sectional study conducted between June 2014 and March 2016.

**Setting** Stand-alone tertiary maternity hospital in an urban setting with approximately 8000 deliveries per year.

**Participants** Women were recruited at their convenience after sonographic confirmation of an ongoing singleton pregnancy (n=502). Detailed dietary and supplement information was available for 398 women. Women <18 years and those who did not speak English fluently were excluded.

**Primary and secondary outcome measures** The differences in dietary micronutrients and macronutrients and maternal folate levels between women who continued to smoke in pregnancy compared with non-smokers.

**Results** Of the 502 women, the mean age was 30.5 (SD 5.6) years, 42.5% were nulliparas, 19.2% were obese and 398 (79.3%) completed the questionnaire satisfactorily. In the 50 (12.6%) current smokers, the micronutrients magnesium, iron, carotene and copper were lower (all p<0.005) whereas sodium and chloride were higher compared with the 348 (87.4%) non-smokers. Smokers reported lower intakes of dietary total folate (p=0.006) compared with non-smokers (i.e., dietary folate equivalents; intake from natural and fortified dietary sources) (p=0.005). Smokers also reported lower intakes of fibre than non-smokers (13.1 g (IQR 7.7) vs 16.3 g (IQR 8.5), p<0.001). The dietary intakes of former smokers compared favourably with non-smokers.

**Conclusions** We found that women who continue to smoke during pregnancy have serious dietary inadequacies which could potentially aggravate fetal growth restriction associated with direct toxicity from cigarettes. This provides a further reason to promote smoking cessation interventions in pregnancy, and highlights the need for dietary and supplementation interventions in women who continue to smoke.

### Strengths and limitations of this study

► This study was a cross-sectional study with a well-characterised study sample.
► The detailed dietary and supplement questionnaires were supervised and analysis of these data included up-to-date brand-level information including details on food fortification with folate.
► Collection of the information in early pregnancy was conducted to minimise the recall bias associated with completion of the questionnaires after delivery.
► A limitation of the study is that convenience recruitment was used rather than consecutive recruitment, as the later was not possible with a single interviewer.
► The information on smoking in this study was self-reported and not validated by biochemical testing. However, any non-disclosers were analysed with the non-smokers and it is likely that the dietary differences we found would be even greater if smoking could be verified by biochemical screening.

## INTRODUCTION

Maternal smoking is strongly associated with an increase in adverse pregnancy outcomes including miscarriage, preterm delivery, congenital malformations and stillbirth.[1 2] In particular, it is associated with an increased risk of intrauterine growth restriction (IUGR) and the risk increases with the number of cigarettes smoked. Smoking is an important modifiable risk factor for IUGR because cessation of smoking in the first half of pregnancy may normalise fetal growth.[3] Yet, many women continue to smoke throughout pregnancy.

Fetal growth and lifelong health outcomes are also influenced by maternal nutrition.[4 5] Deficiencies of both maternal micronutrients and macronutrients have been associated with aberrant fetal growth. In a systematic review and meta-analysis of 62 studies from developed countries, pregnant women had

dietary folate, iron and vitamin D dietary intakes consistently below nutrient recommendations.[6] Forty-one of the studies, however, did not report maternal smoking status which may influence dietary intakes.

This cross-sectional study compared the dietary intakes of micronutrients and macronutrients of self-reported current smokers with the intakes of non-smokers at the first prenatal visit in a large maternity hospital.

## METHODS

Women were recruited at their convenience after sonographic confirmation of an ongoing singleton pregnancy during their first prenatal visit between June 2014 and March 2016. Women with multiple pregnancies or women who did not speak English fluently were excluded. Clinical and sociodemographic details were computerised routinely by a trained midwife using a standard questionnaire and a barcode system. The hospital questionnaire included questions on whether the woman had never smoked, had stopped smoking before her first visit or was continuing to smoke. The number of cigarettes smoked per day was also recorded as 1–5/day, 6–10/day or >10/day. The maternal weight and height were measured by a midwife at the first antenatal visit and the body mass index (BMI) was calculated and categorised. This is more accurate than a self-reported prepregnancy weight.[7] Body weight, BMI and body composition have been shown not to vary significantly across the first trimester in a study of 1000 women.[8] Written informed consent was obtained.

All women were asked to complete a retrospective 4-day diet history (DH) at the first antenatal visit. This was combined with a previously validated Food Frequency Questionnaire (FFQ) which had been customised to capture food fortification with folate at brand level in Ireland.[9] Quantitative food intake data from the DH were divided by four to derive the average daily intake for each food. These data were entered into Nutritics V.3.7 software to convert the food intakes into nutrient measurements. The nutrient requirement reference levels for pregnancy (either average requirement, population reference intake or adequate intake) were obtained from the European Food Safety Authority guideline.[10] Levels for sodium, chloride and vitamin E are not established.

To overcome the possible limitation associated with nutritional surveys of under-reporting, a calculation was applied to identify any likely under-reporters which is based on energy intake (EI), basal metabolic rate (BMR) and physical activity level (PAL).[11 12] Based on the reported PAL, the lowest plausible PAL threshold (from 1.45 to 2.20) was determined. If the EI to BMR ratio fell below the plausible threshold for their PAL then they were identified as a dietary under-reporter. Plausible EI reporters were identified in this way.

Preliminary analyses conducted demonstrated that dietary composition was similar between never smokers and former smokers (online supplementary tables 1 and 2), therefore, analyses combined never and former smokers into one 'non-smokers' group.

Information on socioeconomic status was analysed using questions from the European Union Survey on Income and Living Conditions.[13] All women had a full blood count taken and other biomarkers including serum folate, whole blood folate and red blood cell (RBC) folate levels were obtained at the same visit. Information on birth outcomes was obtained from the hospital-computerised database which is updated immediately after delivery.

Data from the questionnaires were anonymised and entered into Microsoft Excel and exported for statistical analysis using SPSS V.20 (IBM). Continuous data were assessed for normality by visual inspection of the histogram and box-plot and calculation of the kurtosis and skewness. Descriptive statistics were used to describe the characteristics of the study sample. The Kruskal-Wallis test for non-parametric data was used to assess the difference between nutrient intakes in women who were current smokers compared with non-smokers. Tests for statistical significance were performed with the level for significance set at $p < 0.05$.

### Patient and public involvement

The authors were aware of the increased risk of adverse pregnancy outcomes associated with maternal smoking and also the association of smoking with poor dietary quality compared with non-smokers. This was the stimulus for our research question.

The patients were not involved in the study design or the recruitment process, however, the Hospital Research Ethics Committee includes members of the public involved in reviewing the methodology, patient information leaflets and questionnaires.

The results of our study were not directly disseminated to the study population. They will be presented locally to educate staff on our findings with the aim of changing practice to benefit our patients.

## RESULTS

Of the 502 women recruited, 398 (79.3%) completed the dietary questionnaire satisfactorily. The study sample was similar to the hospital population.[14] Women cited time constraints as the main reason for not completing the dietary questionnaire and those who did fill out the dietary questionnaire were more likely to be nulliparous (p=0.012) and Irish-born (p<0.001). Comparing current smokers to non-smokers, they were younger (p<0.001) with a higher percentage of relative income poverty compared with the non-smokers (non-smokers; 21.6% (40/185) vs current smokers; 56.5% (13/23), p=0.009). As expected, the mean birth weight was lower in smokers compared with non-smokers (table 1).

Table 2 presents the dietary intakes overall and stratified by smoking status. The micronutrients magnesium, iron, carotene, copper, vitamin C and riboflavin were lower in smokers compared with non-smokers, but

**Table 1** Comparison of characteristics between smokers and non-smokers with dietary data available (n=398)

| | All women (n=398)* | Smokers (n=50) | Non-smokers (n=348)† | P values |
|---|---|---|---|---|
| Age (years) (mean±SD) | 30.5±5.6 | 27.0±6.44 | 31.0±5.3 | <0.001 |
| BMI (kg/m$^2$) (mean±SD) | 25.9±5.4 | 25.3±4.7 | 26.0±5.5 | 0.418 |
| Obese (%) | 19.2 (76) | 22 (11) | 18.7 (65) | 0.323 |
| Nulliparous (%) | 45.5 (181) | 46.0 (23) | 45.4 (158) | 0.937 |
| Irish-born (%) | 76.9 (306) | 82 (41) | 76.1 (265) | 0.739 |
| Alcohol intake during pregnancy (%) | 4.8 (19) | 6.1 (3) | 4.7 (16) | 0.863 |
| Birth weight (g) (mean±SD) | 3426.9±586.1 | 3134.4±622.1 | 3469.46±569.2 | <0.001 |

Significance level—0.05, statistical tests used— $\chi^2$ test for categorical variables and independent samples t-test for continuous variables).
*Of the 398 women studied, delivery outcomes were available on 378; 330 who were non-smokers and 48 who were current smokers. The remaining women delivered elsewhere.
†As dietary composition was similar between never smokers and former smokers (online supplementary tables 1 and 2), they are combined into one 'non-smokers' group.
BMI, body mass index.

the micronutrients sodium and chloride were higher. Compared with non-smokers, women who smoked had lower levels of dietary folate (p=0.006, table 2) and were less likely to take prepregnancy folic acid (p<0.001, table 3). Women who smoked had a lower fibre intake compared with non-smokers (p<0.001) (table 2).

There was no relationship between the reported number of cigarettes smoked per day (1–5 cigarettes/day compared with ≥6 cigarettes/day) and macronutrient and micronutrient intakes.

Comparing the dietary intakes of former smokers (n=155) with never smokers (n=193), the former smokers had superior intakes of the micronutrients iron, zinc, vitamin B$_{12}$, vitamin B$_2$ (riboflavin) and Vitamin B$_3$ (niacin) (online supplementary tables 1 and 2). They had higher intakes of protein, sodium and chloride.

Table 3 compares the dietary folate intakes for smokers with non-smokers. Non-smokers reported a greater total dietary folate intake (p=0.006) with a superior intake of natural dietary folate compared with smokers (median 193.9 µg (IQR 93.9) vs median 174.1 µg (IQR 112.7) (p=0.029)). However, both groups had a low percentage of women meeting WHO dietary folate equivalent recommended for pregnancy (non-smokers; 6.3% vs smokers; 6.0%) and breastfeeding (non-smokers; 10.9% vs smokers; 6%). This dietary information is supported by the maternal folate measured at the first prenatal visit. Non-smokers had higher serum folate levels (35.2±17.0 nmol/L vs 22.3±19.5 nmol/L, p<0.001) and red cell folate levels (1195.1±440.0 nmol/L vs 820.0±362.5 nmol/L, p<0.001) with 71.3% (248) of the non-smokers achieving the optimal RBC folate level of 906 nmol/L compared with only 38.0% (19) of the smokers.

Of the 348 women in the non-smoking group, 63.8% (222) were plausible EI reporters compared with 68% (34) of the women in the current smoker group. Comparing the dietary intakes including any plausible EI reporters, lower intakes of fibre (p=0.004), iron, copper, magnesium, carotene (all p<0.05) and folic acid (p=0.009) and a

higher chloride intake (p<0.05) remained in the current smokers compared with non-smokers (online supplementary tables 3 and 4).

## DISCUSSION

Based on self-reporting at the first prenatal visit, we found that women who continued to smoke during pregnancy were more likely to have deficiencies of both dietary micronutrients and macronutrients than women who did not smoke. Interestingly, women who had stopped smoking had dietary intakes which compared favourably with women who never smoked.

Maternal smoking is strongly associated with adverse clinical outcomes for both the woman and her offspring and this study highlights that dietary quality and deficiencies are another area of concern. Our findings that smokers have a poor dietary composition relative to non-smokers is consistent with many previous studies, including those among the general adult population[15–18] and pregnant women.[19 20]

Data from 11 260 adults aged 19–74 years from the second National Health and Nutrition Examination Survey were analysed to determine food and nutrient differences between smokers and non-smokers.[21] In smokers, there was a lower intake of vitamins A and C, folate and fibre and the intakes decreased as cigarette consumption increased. Smokers were less likely to consume fruit and vegetables, high fibre grains, low-fat milk as well as vitamin and mineral supplements. These findings raised concerns that the lifelong risk of cancer was increased by cigarette smoking and by dietary deficiencies.

A Swiss population study of 2301 men and 2306 women reported that current heavy smokers consumed less total vegetable proteins, carbohydrates, fibre, betacarotene, fruit and vegetables compared with non-smokers (p values all significant).[22] They also drank more alcohol and coffee (p<0.0001 and p<0.005, respectively). The female smokers also consumed less complex carbohydrates (p<0.002) and

Table 2  Daily macronutrient and micronutrient intakes in smokers compared with non-smokers

|  | Smokers  (n=50) | Non-smokers (n=348)* | P values |
|---|---|---|---|
| Protein (g) | 69.1 (31.3) | 74.4 (28.0) | 0.168 |
| Carbohydrate (g) | 204.3 (95.7) | 198.8 (73.7) | 0.820 |
| Fat (g) | 75.5 (42.8) | 72.7 (39.1) | 0.421 |
| Saturates (g) | 30.3 (15.1) | 27.2 (15.7) | 0.353 |
| Monounsaturated fat (g) | 25.3 (13.7) | 24.5 (13.6) | 0.928 |
| Polyunsaturated fat (g) | 8.8 (7.0) | 10.3 (7.0) | 0.176 |
| Fibre (g) | 13.1 (7.7) | 16.3 (8.5) | <0.001 |
| Sodium (mg) | 2100.7 (1158.7) | 1877.0 (971.9) | 0.095 |
| Potassium (mg) | 2371.2 (1085.7) | 2533.1 (1038.5) | 0.305 |
| Calcium (mg) | 797.8 (538.06) | 788.4 (536.4) | 0.601 |
| Magnesium (mg) | 196.0 (75.8) | 226.9 (100.5) | 0.003 |
| Phosphorous (mg) | 1060.5 (381.7) | 1180.8 (490.1) | 0.063 |
| Iron (mg) | 7.9 (3.9) | 9.5 (4.25) | 0.002 |
| Copper (mg) | 0.7 (0.4) | 0.9 (0.5) | 0.001 |
| Zinc (mg) | 7.1 (3.9) | 7.8 (3.7) | 0.227 |
| Chloride (mg) | 3187.3 (1678.0) | 2843.7 (1345.8) | 0.067 |
| Iodine (µg) | 91.5 (89.5) | 104.5 (77.0) | 0.500 |
| Retinol (µg) | 312.6 (225.0) | 282.9 (242.3) | 0.916 |
| Carotene (µg) | 1488.3 (3427.4) | 3213.2 (4479.5) | 0.004 |
| Vitamin D (µg) | 2.3 (2.5) | 2.4 (2.6) | 0.547 |
| Vitamin C (mg) | 71.6 (74.3) | 83.8 (83.0) | 0.035 |
| Vitamin E (µg) | 7.1 (5.1) | 8.2 (5.6) | 0.196 |
| Thiamine (mg) | 1.3 (0.6) | 1.4 (0.7) | 0.100 |
| Riboflavin (mg) | 1.3 (0.9) | 1.4 (0.7) | 0.040 |
| Niacin (mg) | 30.5 (16.8) | 34.1 (14.4) | 0.089 |
| Vitamin $B_6$ (mg) | 1.7 (1.0) | 1.9 (0.9) | 0.451 |
| Vitamin $B_{12}$ (µg) | 4.0 (3.5) | 4.0 (2.6) | 0.767 |
| Folate (natural and fortified) (µg) | 202.1 (127.3) | 241.9 (142.8) | 0.006 |

Significance level—0.05, statistical tests used; Kruskal-Wallis.

*As dietary composition was similar between never smokers and former smokers (online supplementary tables 1 and 2), they are combined into one 'non-smokers' group.

less iron (p<0.02). As in this study, the diet of ex-smokers was similar to never smokers. This demonstrates that it is likely that quitting smoking is associated with other positive lifestyle modifications including improvement of diet.

A study comparing 60 smokers to 80 non-smokers in pregnancy demonstrated the oxidative stress associated with smoking through elevated levels of the oxidative malondialdehyde and corresponding lower levels of known antioxidants such as vitamin E, vitamin A and beta carotene in the plasma of smokers and cord blood of their infants compared with non-smokers and their infants.[23] Smoking produces free radicals that deplete antioxidant levels and these effects could be amplified by poor dietary intake of these important micronutrients.

The reasons for the differences in the dietary intakes between smokers and non-smokers are generally unexplained. One factor that may contribute is the impact of smoking on taste perception. A study conducted with 83 smokers and 48 non-smokers (male and female) using electrogustometry detected taste disturbance in the smokers that was dependent on the intensity of smoking. A follow-up of 24 of the smokers who quit smoking demonstrated a rapid recovery in taste sensitivity within 2 weeks in certain loci of the tongue and up to 12 months at other sites.[24] The daily requirement for many micronutrients increases in pregnancy to meet the higher physiologic demands.[25] There are both short-term and long-term consequences of fetal undernutrition. A Norwegian prospective cohort study of 66 000 pregnant women conducted in 2002–2008 used a FFQ to assess dietary habits and established three patterns: 'prudent' (vegetables, fruits, oils, water as beverage, whole grain cereals and fibre rich bread), 'Western' (salty and sweet snacks, white bread, desserts and processed meats) and

**Table 3** Total median (IQR) maternal dietary folate (including fortified folic acid) per day in early pregnancy and haematological measures of folate in non-smokers compared with current smokers (n=398)

| | Non-smokers (n=348)* | Smokers (n=50) | P values |
|---|---|---|---|
| Natural dietary folate intake (µg) | 193.9 (93.9) | 174.1 (112.7) | 0.029 |
| Fortified dietary folate intake (µg) | 33.8 (50.7) | 16.9 (59.4) | 0.076 |
| Total dietary folate intake (µg) | 241.9 (142.8) | 202.1 (127.1) | 0.006 |
| Dietary folate equivalent (µg) | 268.2 (190.7) | 214.6 (146.9) | 0.005 |
| Achieving WHO DFE for pregnancy (≥600 µg DFE) (%, n) | 6.3 (22) | 6.0 (3) | 0.930 |
| Achieving WHO DFE for lactation (≥500 µg DFE) (%, n) | 10.9 (38) | 6.0 (3) | 0.285 |
| Prepregnancy FA use (%, n) | 48.0 (167) | 10.0 (5) | 0.015 |
| Serum folate (nmol/L), (median, IQR) | 35.2±17.0 | 22.3±19.5 | <0.001 |
| Red blood cell (RBC) folate (nmol/L), (mean, SD) | 1195.1±440.0 | 820.0±362.5 | <0.001 |
| Achieving optimal RBC folate (%, n)† | 71.3 (248) | 38.0 (19) | <0.001 |

Significance level—0.05, statistical tests used—$\chi^2$ test for categorical variables and independent samples t-test for continuous variables (normally distributed) and Kruskal-Wallis for comparison of non-normally distributed variables).

DFE—dietary folate equivalent—for natural folate from food—1 µg folate from food is equal to 1 µg DFE. For food fortified with folate 1 µg of folate=1.7 DFE, as natural folate from food is 50% bioavailable compared with 85% bioavailability from fortified folate and folate supplementation (85/50=1.7).

*As dietary composition was similar between never smokers and former smokers (online supplementary tables 1 and 2), they are combined into one 'non-smokers' group.

†The reported optimal threshold concentration for RBC folate is reported as 906 nmol/L by Daly et al.[30] At and above this concentration, the risk of neural tube defects (NTDs) was reduced to <8 NTDs/10 000 live births.

FA, folic acid.

'traditional' (potatoes, fish). After adjustment for covariates, higher scores for the 'prudent' dietary choices were associated with a decreased risk of preterm delivery (0.88, 95% CI 0.80 to 0.97). However, there was no independent association with the 'Western' diet pattern and preterm delivery. It was concluded that low adherence to the 'prudent' pattern is a stronger indicator of unhealthy diet than intake of the Western diet.[26]

Maternal smoking is associated with higher rates of small for gestational age infants, but fetal growth retardation can extend beyond that cut-off point, for example, when an infant born at 3 kg fails to reach their growth potential of 3.5 kg. Fetal growth is dependent on their nutrient uptake and the start point of this process is the maternal intake. A study based in India from 1994 to 1997 of 797 pregnant women studied maternal diet and neonatal outcomes. There was a strong correlation between consumption of green leafy vegetables (GLVs), milk and fruit with a higher birth weight. For GLVs, it was shown that women who consumed them on alternate days delivered infants almost 200 g heavier than those who never ate them. Furthermore, there were correlations between serum vitamin C and RBC folate levels and birth weight.[5] Considering the long-term consequences of undernutrition, the implications of this on the fetus are described by Barkers et al hypothesis with an increase in type 2 diabetes mellitus, hypertension and coronary heart disease.[4 27]

A Canadian study conducted from 2001 to 2007 of 1646 self-reported smokers compared with 19 292 non-smokers, reported an increase in adverse perinatal and neonatal outcomes among smokers. There was an increased risk of preterm delivery (22.2% vs 12.4%, p<0.05), intrauterine fetal demise (OR 2.4 (95% CI 1.4 to 4.2), a mean birth weight approximately 200 g lower in the current smokers (p<0.05) and an increased risk of neonatal death (1.2% vs 0.6%, p<0.05).[28]

A systematic review and meta-analysis of 13 articles found a pooled OR of neural tube defects (NTDs) in offspring of smokers of 1.03 (95% CI 0.80 to 1.33). When subgroup analysis was performed based on geographical region, the NTD risk was 1.39 (95% CI 1.18 to 1.64) in Europe and 0.88 (95% CI 0.66 to 1.17) in the USA.[29] The transatlantic difference may reflect that the diet of smokers in Europe is less likely to be fortified with folate where fortification is voluntary and not mandatory. Analysis by NTD type also revealed a relationship between maternal smoking and spina bifida (OR 1.55 (95% CI 1.06 to 2.26)).[24] Therefore, inadequate dietary folate and decreased use of supplementation may contribute to the slight increased risk of NTDs in smokers. The results of our study demonstrated that current smokers were less likely than non-smokers to achieve the optimal RBC folate level of 906 nmol/L associated with prevention of NTDs. Above this optimal RBC folate concentration, the risk of NTDs is reported to reduce to <8/10 000 live births.[31]

Women who continue to smoke in pregnancy may benefit from dietary intervention. A study of 862 women and infant pairs assessed maternal dietary quality at 24–28 weeks gestation. Lower dietary quality was associated with lower educational attainment, maternal smoking, obesity (prepregnancy) and a lack of exercise during pregnancy.

It was found that increased dietary quality was associated with a reduced likelihood of a small for gestational age infant. Furthermore, a higher diet quality was positively associated with birth weight among former and current smokers.[32]

A study of two mother and infant cohorts (2461 pairs in Spain and 889 pairs in Greece) found that a high Mediterranean diet (MD—vegetables, legumes, fruit, nuts, cereals, seafood and fish and dairy products) adherence was associated with a lower risk of a growth restricted infant (risk ratio 0.5, 95% CI 0.3 to 0.9). When the analysis was stratified according to smoking status, higher MD adherence was associated with higher birth weight and birth length in the infants of mothers who continued to smoke.[33]

A strength of our study was that it was conducted prospectively and the study sample was well characterised. The detailed questionnaires on diet and supplements were supervised and analysis of these data considered up-to-date brand-level information, including details on food fortification. The information was collected in early pregnancy to minimise the recall bias associated with the completion of questionnaires after childbirth.

A limitation of the study was that convenience recruitment was used rather than consecutive recruitment which is not feasible in a busy maternity service with a single interviewer. The information on smoking was self-reported and not validated by biochemical testing. However, any non-disclosures of smoking were analysed with the non-smokers and thus, it is likely that the dietary differences we found would be even greater if smoking could be verified by biochemical screening. Dietary information was gathered in the first trimester only and it is not known if dietary choices changed as pregnancy advanced. An Irish study of 285 women who completed a 3-day food diary in each trimester used cluster analysis to identify two major dietary patterns, 'unhealthy' and 'health conscious'. Comparing the energy and nutrient intakes from the first trimester to the third trimester, they found no significant change in either cluster.[34] It is also difficult to compare our findings on the relationship between dietary intakes and smoking behaviour in our study with previous reports because surveys on dietary intakes in adults usually excluded women who were pregnant or breast feeding.[21]

## CONCLUSIONS

Our findings that pregnant women who continue to smoke have dietary inadequacies strengthens the public health case for screening women for smoking in early pregnancy and for offering support for smoking cessation. The dietary intakes of former smokers suggest that dietary intakes improve after smoking cessation. For those women that continue to smoke in pregnancy, it also highlights the need to advise women about healthy eating behaviours and about the need to start folic acid supplementation.

**Acknowledgements** We thank the Friends of the Coombe for their support. We thank the women who participated and our laboratory and midwifery colleagues for their assistance.

**Contributors** EOM contributed to the conception and design of the study, collected data, performed data analysis and wrote and edited the manuscript. SC conducted the primary study involving patient recruitment, issuing questionnaires, analysis of food diaries and gathering data and contributed to the conception and design of this study and drafting of the manuscript. RAKK and CMER were involved in the conception and design of this study, data analysis and contributed to writing and editing. AM was involved in conception and design of the study, drafting sections of the manuscript and approving the final submitted version. MJT contributed to the conception and design of the study, analysis of data and writing and editing of the manuscript.

**Funding** This work was supported by the all island body Safefood (grant number 02-2015).

**Competing interests** None declared.

**Patient consent** Obtained.

**Ethics approval** The study was approved by the Coombe Women and Infants University Hospital Research and Ethics Committee (REC)–reference number 27-2013.

**Provenance and peer review** Not commissioned; externally peer reviewed.

**Data sharing statement** Extra data are available by emailing eimer.om@gmail.com.

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
