## [Reviewer comments · BMJ Open]

ARTICLE DETAILS

TITLE (PROVISIONAL)	Comparison at the first prenatal visit of the maternal dietary intakes of smokers with non-smokers in a large maternity hospital: a cross sectional study.
AUTHORS	O'Malley, Eimer; Cawley, Shona; Reynolds, Ciara; Kennedy, Rachel; Molloy, Anne; Turner, Michael

VERSION 1 – REVIEW

REVIEWER	Monica Tarcea University of Medicine and Pharmacy Tirgu Mures
REVIEW RETURNED	05-Feb-2018

GENERAL COMMENTS	Dear authors, I read your manuscript with interest, its a challeging topic, very important for the preventive domain of healthcare. You have a lot of data, validated questionnaire, good statistics, proper discussions and useful conclusion, but you have to rewrite the Methodology and Results sections, and to be more specific and clear in relation between the Title of the article, the objective and the outcomes and conclusion. You have too much tables, just add tabel 2 with the supplement of it, erase Tabel 1 (is not necessary for your article to evaluate the women without dietary date) and explain more data from Supplementary table 4. It is needed to review the Title: If you are evaluating the maternal dietary intake at the first prenatal visit, you cant use data from the period after delivery and also you cant evaluate if the pregnant women were smoking during all pregnancy. I suggest to erase the part from the title "at the first prenatalal visit", or to add "from the first prenatal visit to delivery". or to replace that part with "during pregnancy". The abstract is not written in the clasical way (Introduction/Background, Material and Methods, Results and Conclusion), but if these are the requiremenst of the journal, its ok. Anyway from my point of view its a confusin withh the groups you chose for the study, it is not interesting for this study to monitore the pregnant women who did not answer properly to diet questionnaire, so focus your data on those 398 and forget about 502. Also, if you just want to evaluate the maternal dietary intakes its not needed to talk about the lowbirth weight newborns, otherwise you can add in title "Compariosn of the maternal dietary intakes of smokers towrads non-smokers and impact on perinatal outcomes / fetal growth". To the Methods section: it is confusing how many questionnaire you used, exactly when in time (related to pregnancy) and on how many women. You said on the second paragra[h that "all women were asked to complete a retrospective foud days diet hostroy before they left the clinic". Before that you said "birth outcomes were
---

	computerized immediately after delivery". Please provide exactly when you asked the women to fill in the DH, at their first prenatal visit or after delivery and when the FFQ was completed/ There were done at different times or one after another? And why two types of questionnaire? If the DH questionnaire was done after deliverance its not ok, because the last four days the women stayed in hospital so the menu was different. Also another issue is when you have registered their weight and height? Its ok to put them in the obese category if they were measured before being pregnant not afterwards...because their weight is changing in the last trimester. Be more specific with the questionnaires and collecting data activities. Also in the first trimester the menu can be changed if they had nausea and vomiting reactions. Also, if you made the biochemical test at the first visit only, it doesnt mean you were monitoring the impact of their dietary intake during all pregnancy period of time...be more specific. The Results also has to be adjusted, you made too many tables. Forget about 502 women, and focused on those 398. It will be better to add to one table three groups: Non-smokers (meaning women who never smoke) + Ex-smokers or Former smokers (women who stop smoking at least one year before pregnancy or when they found out they are pregnant) + smokers during pregnancy (you have already the data calculated). If you made the questionnaire at their first prenatal visit and asked them if they are smokers and they said they are not any more but they stopped recently smoking (just one week or one month before smoking) the impact is different from those who never smoked or ceased smoking at least one year before. If you already separate never smokers, ex-smokers and smokers...use all three groups in one table, it will be more validating. Please explain more what you mean by "plausible energy intake reporters"and supplement table no 3 and 4. Discussions and Conclusion are OK. I recommend your manuscript to be published after major revision.
--	--

REVIEWER	Jennifer A. Emond Dartmouth College, USA
REVIEW RETURNED	19-Feb-2018

GENERAL COMMENTS	This study compared the dietary composition during the first trimester of pregnancy between women who smoked and did not smoke among a sample of healthy women with singleton pregnancies. The study is a useful descriptive analysis that can speak to the magnitude of the differences in diet quality by smoking status. Overall, the presentation of the findings and Discussion can be refined, and there are a few methodological issues that need to be addressed (noted below). Overall: Consider incorporating two key studies in this area regarding dietary intake and smoking during pregnancy and fetal growth. Better Diet Quality during Pregnancy Is Associated with a Reduced Likelihood of an Infant Born Small for Gestational Age: An Analysis of the Prospective New Hampshire Birth Cohort Study. Emond JA, Karagas MR, Baker ER, Gilbert-Diamond D. J Nutr. 2018 Jan 1;148(1):22-30. Mediterranean diet adherence during pregnancy and fetal growth: INMA (Spain) and RHEA (Greece) mother-child cohort studies.
--

Chatzi L, Mendez M, Garcia R, Roumeliotaki T, Ibarluzea J, Tardón A, Amiano P, Lertxundi A, Iñiguez C, Vioque J, Kogevinas M, Sunyer J; INMA and RHEA study groups. Br J Nutr. 2012 Jan;107(1):135-45.

Abstract

Well summarized

Is the abstract supposed to be an outline with lower case titles?

Strengths and Limitations

As analyzed, this study is not prospective. See my comment below about dropping infant birth outcomes.

I wouldn't consider a convenience sample a study strength.

Introduction

Well written.

Lines 29-31: "Forty-one of the studies, however, did not report maternal smoking status." It is not clear to the reader why that is important - are you specifically addressing that there is a limited understanding for how dietary intakes may differ by smoking status?

Methods

This study is not a prospective study. It is cross-sectional. It is comparing dietary intake between smokers and non-smokers at one time point. That is the primary research question.

Page 4, lines 50-51: "Women who did not understand English were excluded." That phrasing is unclear. Was it that women who didn't speak English fluently were excluded, or were there other cognitive reasons for exclusion?

Page 5, line 6: Was BMI computed for pre-pregnancy weight? Were those values self-reported?

Page 5, line 10-12: Do not include birth outcomes in this analysis. They are not part of the primary research question (specifically, does dietary composition during pregnancy differ between smokers and non-smokers?). Also, by including those outcomes, you make the reader question why you didn't do a more extensive analysis looking at dietary quality and smoking status on birth outcomes.

Page 6, line 5: "Tests for statistical significance were performed at the 95% level (i.e., $p < 0.05$)". That is incorrect – 0.05 is the level of statistical significance. 95% is the margin of error.

The Methods needs to include the details about comparing dietary intakes to WHO reference levels. Or, move that comparison out of the Results and into the Discussion.

The Methods needs to include the details on how "plausible energy intake reporters" were defined (as reported in the Results).

Results

	When discussing the sample – the sample is the study *sample*, not the study population. For example, page 6, line 18: “The characteristics of the study the study population is shown...”. That should be written as: “The characteristics of the study the study *sample* *are* shown...”. Line 38: “Table 3 shows theand analyses the...” I think that should read “analyzes”. However, the table doesn’t analyze the data. Thus, I’d rewrite as “Table 3 presents the dietary intakes overall and stratified by smoking status.” Page 7, lines 2-7: “ There was no relationship”. Drop the line “This may have been due to a small sample size.” That is not relevant here (move that to the Discussion if needed). However, why do you think diet would differ with smoking intensity? I would consider dropping that aspect of the analysis. Page 7: When discussing Supplementary Tables 1 and 2, the reference to combining never and former smokers is confusing. Do all of the following analyses combine those two groups? If they do – make sure each table has a footnote to clarify that, and move that detail to the Methods. In fact, you can state in the Methods that because preliminary analyses documented that dietary composition was similar between never smokers and former smokers (Supplemental table X), analyses combined never and former smokers into one “non-smokers” group. In all tables, include the tests used for statistical significance as a footnote (i.e., define where the P-values came from). Discussion Page 8, lines 26-44. These analyses looked at what women ate. It does not look at “availability” of nutrients (which implies bioavailability). Thus, quitting smoking does not impact the availability of any nutrient. Likely, women quit smoking and made other positive health changes – including improving their diets. Move the discussion of strengths and limitations to the penultimate paragraph. This discussion includes an excessive discussion of past studies which have reported differences in dietary intakes among smokers and non-smokers. This information (across multiple paragraphs) can be succinctly summarized with a brief statement such as “Our findings that smokers have a poor dietary composition relative to non-smokers is consistent with many previous studies, including those among pregnant women (references) and the general adult population (references).” The Discussion needs to include some discussion of how smoking is correlated with other poor health behaviors, including poor dietary habits. Specifically, please recognize that it is not quitting smoking that changes a woman’s dietary intake, but other lifestyle changes that may covary with quitting smoking. You could discuss the importance of quality nutrition during pregnancy, and the implications on maternal and infant health. Then comment on the dangers of smoking during pregnancy – and then hypothesize about the increased risks women may face if they smoke and also have a poor diet quality.
--	--

	It is also possible that women who smoke during pregnancy may benefit the most from a dietary intervention (see Emond et al, Chatzi et al), and you noted in the Conclusions. Include some mention as to whether you think dietary intake (as measured at the 1st trimester) is likely to be consistent throughout pregnancy. The Conclusions paragraph is well written.
--	---

VERSION 1 – AUTHOR RESPONSE

Point by point response

Reviewers' Comments to Author:

Reviewer: 1

Reviewer Name: Monica Tarcea

Institution and Country: University of Medicine and Pharmacy Tirgu Mures

Please state any competing interests or state 'None declared': No conflict of interest.

Dear authors,

I read your manuscript with interest, its a challeging topic, very important for the preventive domain of healtcare. You have a lot of data, validated questionnaire, good statistics, proper discussions and useful conclusion, but you have to rewrite the Methodology and Results sections, and to be more specific and clear in relation between the Title of the article, the objective and the outcomes and conclusion.

Reviewer 1 comment

1. You have too much tables, just add tabel 2 with the supplement of it, erase Tabel 1 (is not necessary for your article to evaluate the women without dietary date) and explain more data from Supplementary table 4.

Response to Reviewer 1

Table 1 compares the general characteristics of the entire study sample (502 women were approached and had phlebotomy to assess serum and red blood cell folate). Only 398 women filled out the dietary questionnaires adequately so we compared the characteristics of the women who provided adequate dietary data to those who did not as we wanted to ensure that these women did

not vary significantly. We found that the women who completed the dietary questionnaire were more likely to be Irish and nulliparous. We feel that it is important to identify differences in the characteristics between those who did complete vs did not complete the questionnaire.

Regarding the explanation of Supplementary Table 4; The methodology used to identify plausible energy intake reporters and exclude likely dietary under-reporters has been elaborated on in the methods on page 6 line 42-55. (The in-text references number 11 (Goldberg et al) and 12 (Black) refer to this methodology). (As advised by reviewer 2 also)

“To overcome the possible limitation associated with nutritional surveys of under-reporting, a calculation was applied to identify any likely under-reporters which is based on energy intake, basal metabolic rate and physical activity level.[8,9] Based on the reported physical activity level (PAL) on a six point scale, the lowest plausible PAL threshold (from 1.45-2.20) was determined. If the energy intake (EI) to basal metabolic rate (BMR) ratio fell below the plausible threshold for their PAL then they were identified as a dietary under-reporter. Plausible energy intake reporters could be identified in this way.”

The results of Supplementary tables 3 and 4 regarding the macronutrient and micronutrient intakes of the plausible reporters only stratified by smoking status are discussed on page 9 line 29-40.

“Of the 348 women in the non-smoking group, 63.8% (222) were plausible energy intake reporters compared to 68% (34) of the women in the current smoker group. Comparing the dietary intakes including any plausible energy intake reporters, lower intakes of fibre ($p=0.004$), iron, copper, magnesium, carotene (all $p<0.05$) and folic acid ($p=0.009$) and a higher chloride intake ($p<0.05$) remained in the current smokers compared with non-smokers (Supplementary Tables 3 and 4).”

Reviewer 1 comment

2.It is needed to review the Title: If you are evaluating the maternal dietary intake at the first prenatal visit, you cant use data from the period after delivery and also you cant evaluate if the pregnant women were smoking during all pregnancy. I suggest to erase the part from the title "at the first

prenatal visit", or to add "from the first prenatal visit to delivery". or to replace that part with "during pregnancy".

Response to Reviewer 1

The title has been changed as advised by the Editor and reviewer 2;

“Comparison at the first prenatal visit of the maternal dietary intakes of smokers with non-smokers in a large maternity hospital: a cross sectional study.”

Reviewer 1 comment

3.The abstract is not written in the classical way (Introduction/Background, Material and Methods, Results and Conclusion), but if these are the requirements of the journal, it's ok. Anyway from my point of view it's a confusion with the groups you chose for the study, it is not interesting for this study to monitor the pregnant women who did not answer properly to diet questionnaire, so focus your data on those 398 and forget about 502. Also, if you just want to evaluate the maternal dietary intakes it's not needed to talk about the lowbirth weight newborns, otherwise you can add in title "Comparison of the maternal dietary intakes of smokers towards non-smokers and impact on perinatal outcomes / fetal growth".

Response to Reviewer 1

Regarding the style of the abstract. This conforms with the requirements of BMJ Open.

Regarding the mention of the study sample of 502 women. This is discussed in response to point 1.

The reviewer comment regarding the inclusion of birth weight outcomes is addressed below as mentioned by reviewer 2 also.

The infant birth weight is significantly different when comparing non-smokers and current smokers.

We feel that this is an important secondary outcome of the research. This paper highlights the dietary inadequacies of smokers and strengthens the case for screening women in early pregnancy and offering smoking cessation support. An increased birth weight is an important clinical outcome associated with non-smoking status compared to infants of current smokers and further strengthens the argument for concentrating efforts in antenatal care to identify smokers and support smoking cessation.

There are many other birth outcomes included in the tables (such as gestational age at delivery, preterm birth rate, rate of small for gestational age and caesarean section rate.) There were no significant differences in these when comparing smokers and non-smokers. They have been deleted from Tables 1 and 2 to simplify the presentation of the data.

Reviewer 1 comment

4. To the Methods section: it is confusing how many questionnaire you used, exactly when in time (related to pregnancy) and on how many women. You said on the second paragra[h that "all women were asked to complete a retrospective foud days diet hostroy before they left the clinic". Before that you said "birth outcomes were cpmputerized immediately after delivery". Please provide exactly when you asked the women to fill in the DH, at their first prenatal visit or after delivery and when the FFQ was completed/ There were doen at different times or one after another? And why two types of questionnaire? If the DH questionnaire was done after deliverance its not ok, because the last four days the women stayed in hospital so the menu was different.

Response to Reviewer 1

Thank you for your comment. The questionnaire used to assess the dietary intake was a four-day retrospective dietary history filled in at the first prenatal visit with a customised food frequency questionnaire used to assess the frequency of consumption of food sources known to be fortified with folic acid. The questionnaire included an assessment of physical activity level so that plausible energy reporters could be differentiated from likely energy under-reporters. None of the patients were in-patients in the hospital so their diet was not influenced by this.

Clinical and socio-demographic information from the interview by the midwife at the first visit and birth outcomes (updated at the time of delivery) were obtained from the hospital computerized database. The methods have been updated to clarify this.

On page 6 line 20-27

“All women were asked to complete a retrospective four-day diet history (DH) at the first antenatal visit. This was combined with a previously validated food frequency questionnaire (FFQ) which had been customized to capture food fortification with folate at brand level in Ireland.[9]”

And on page 7 line 16-20

“Information on birth outcomes were obtained from the hospital computerized database which is updated immediately after delivery.”

Reviewer 1 comment

5. Also another issue is when you have registered the weight and height? Its ok to put them in the obese category if they were measured before being pregnant not afterwards...because their weight is changing in the last trimester. Be more specific with the questionnaires and collecting data activities.

Response to Reviewer 1

This is also addressed in the comments from reviewer 2

The BMI was calculated based on a height and weight measured by the midwife at the first antenatal clinic visit. This is considered more accurate than self-reported values. The pre-pregnancy weight was not obtained. This would not be feasible and previous studies have shown that self-reported weights are inaccurate [7]. A study conducted previously by our research group studied the weight, calculated BMI and body composition of 1000 women in the first trimester of pregnancy. The mean weight, BMI and body composition measures did not differ when the women were stratified according to number of weeks gestation from 5-13 weeks [8]. These references are included in the updated manuscript.

The text in the methods was updated as follows (page 6 line 7-18);

“The maternal weight and height were measured by a midwife at the first antenatal visit and the Body Mass Index (BMI) was calculated and categorized. This is more accurate than a self-reported pre-pregnancy weight. Body weight, BMI and body composition have been shown to not vary significantly across the first trimester in a study of 1000 women.”

Reviewer 1 comment

6. Also in the first trimester the menu can be changed if they had nausea and vomiting reactions. Also, if you made the biochemical test at the first visit only, it doesnt mean you were monitoring the impact of their dietary intake during all pregnancy period of time...be more specific.

Response to Reviewer 1

All interactions with the women in the study sample were at the first antenatal visit (dietary questionnaires and the blood tests).

Reviewer 2 also raises the comment whether the diet may vary according to trimester. This is addressed in the discussion.

We agree that the first trimester diet may be influenced by nausea and vomiting compared to the second and third trimester. There are limited studies in the literature that have assessed the dietary quality of the same cohort of pregnant women in all three trimesters. The discussion has been updated to reference a study that demonstrated the dietary intake of the same cohort of women in the first trimester did not change significantly in the second and third trimester. On page 14 line 38-56; “Dietary information was gathered in the first trimester only and it is not known if dietary choices changed as pregnancy advanced. An Irish study of 285 women who completed a three-day food diary in each trimester used cluster analysis to identify two major dietary patterns, “Unhealthy” and “Health conscious”. Comparing the energy and nutrient intakes from the first trimester to the third trimester they found no significant change in either cluster.” [33]

Reviewer 1 comment

7. The Results also has to be adjusted, you made too many tables. Forget about 502 women, and focused on those 398. It will be better to add to one table three groups: Non-smokers (meaning women who never smoke) + Ex-smokers or Former smokers (women who stop smoking at least one year before pregnancy or when they found out they are pregnant) + smokers during pregnancy (you have already the data calculated). If you made the questionnaire at their first prenatal visit and asked them if they are smokers and they said they are not any more but they stopped recently smoking (just one week or one month before smoking) the impact is different from those who never smoked or ceased smoking at least one year before. If you already separate never smokers, ex-smokers and smokers...use all three groups in one table, it will be more validating.

Response to Reviewer 1

Supplementary tables 1-4 are meant only for clarification and provided as additional material for inclusion online. Supplementary tables 1 and 2 show that the dietary intakes of former smokers

compare favourably and in some instances are improved compared to never smokers. This provides justification for their inclusion as one group “non-smokers” in Tables 2,3 and 4.

Supplementary tables 3 and 4 demonstrate the macronutrient and micronutrient intakes of those women who were deemed to be Plausible energy reporters as the inclusion of dietary under-reporters can bias the results. We found that the analyses were largely unchanged with respect to the differences in intakes between smokers and non-smokers so the dietary data for all 398 women is presented in the main Tables 3 and 4 and the supplementary data provided to verify this.

Reviewer 1 comment

8.Please explain more what you mean by "plausible energy intake reporters"and supplement table no 3 and 4.

Response to Reviewer 1

Thank you for this comment. As requested by reviewer 2 also, plausible energy intake is explained in more detail in the methods.

Page 6 line 42-55

“To overcome the possible limitation associated with nutritional surveys of under-reporting, a calculation was applied to identify any likely under-reporters which is based on energy intake, basal metabolic rate and physical activity level.[11,12] Based on the reported physical activity level (PAL), the lowest plausible PAL threshold (from 1.45-2.20) was determined. If the energy intake (EI) to basal metabolic rate (BMR) ratio fell below the plausible threshold for their PAL then they were identified as a dietary under-reporter. Plausible energy intake reporters could be identified in this way.”

Reviewer 1 comment

Discussions and Conclusion are OK.

I recommend your manuscript to be published after major revision.

Reviewer: 2

Reviewer Name: Jennifer A. Emond

Institution and Country: Dartmouth College, USA

Please state any competing interests or state 'None declared': None

This study compared the dietary composition during the first trimester of pregnancy between women who smoked and did not smoke among a sample of healthy women with singleton pregnancies. The study is a useful descriptive analysis that can speak to the magnitude of the differences in diet quality by smoking status. Overall, the presentation of the findings and Discussion can be refined, and there are a few methodological issues that need to be addressed (noted below).

1. Overall:

Consider incorporating two key studies in this area regarding dietary intake and smoking during pregnancy and fetal growth.

Better Diet Quality during Pregnancy Is Associated with a Reduced Likelihood of an Infant Born Small for Gestational Age: An Analysis of the Prospective New Hampshire Birth Cohort Study.

Emond JA, Karagas MR, Baker ER, Gilbert-Diamond D.

J Nutr. 2018 Jan 1;148(1):22-30.

Mediterranean diet adherence during pregnancy and fetal growth: INMA (Spain) and RHEA (Greece) mother-child cohort studies.

Chatzi L, Mendez M, Garcia R, Roumeliotaki T, Ibarluzea J, Tardón A, Amiano P, Lertxundi A, Iñiguez C, Vioque J, Kogevinas M, Sunyer J; INMA and RHEA study groups.

Br J Nutr. 2012 Jan;107(1):135-45.

2. Abstract

Well summarized

Is the abstract supposed to be an outline with lower case titles?

Response to Reviewer 2

The headings of the abstract (Objectives, Design, Setting etc.) have been amended and the subtitles are now capitalised.

3 Strengths and Limitations

Reviewer 2 comment

As analyzed, this study is not prospective. See my comment below about dropping infant birth outcomes.

Response to Reviewer 2

The study type was change cross-sectional and as requested by the editor incorporated into the title. Please see response to point re Page 5, line 10-12 below regarding infant birth outcomes

Reviewer 2 comment

I wouldn't consider a convenience sample a study strength.

Response to Reviewer 2

Thank you for this comment. The point of convenience recruitment was made as a limitation of the study compared to consecutive recruitment. This is clarified in the Strength and Limitations section (page 4 line 20-24).

"A limitation of the study is that convenience recruitment was used rather than consecutive recruitment, as the later was not possible with a single interviewer. "

4. Introduction

Reviewer 2 comment.

Well written.

Lines 29-31: "Forty-one of the studies, however, did not report maternal smoking status." It is not clear to the reader why that is important - are you specifically addressing that there is a limited understanding for how dietary intakes may differ by smoking status?

Response to Reviewer 2

Thank you for this constructive comment. The intention was to highlight that the dietary intakes of folate, iron and vitamin D were below the recommended values in pregnancy in the systematic review and meta-analysis cited. However, 41 of the 62 studies did not report the maternal smoking status. Based on our findings, the dietary intake differs by smoking status and the point is made to show that consideration of smoking status is an important factor when assessing dietary intakes. The sentence is amended to demonstrate this as follows (page 5, line 29-33)

"Forty-one of the studies, however, did not report maternal smoking status which may influence dietary intakes."

5. Methods

Reviewer 2 comment.

This study is not a prospective study. It is cross-sectional. It is comparing dietary intake between smokers and non-smokers at one time point. That is the primary research question.

Response to Reviewer 2

The study type has been updated to cross-sectional at all instances in the text.

Reviewer 2 comment.

Page 4, lines 50-51: "Women who did not understand English were excluded." That phrasing is unclear. Was it that women who didn't speak English fluently were excluded, or were there other cognitive reasons for exclusion?

Response to Reviewer 2

Thank you for this constructive comment. Women who didn't speak English fluently were excluded.

This had been updated in the methods section on page 5, line 50-53 as follows;

“ women who did not speak English fluently were excluded”

Reviewer 2 comment.

Page 5, line 6: Was BMI computed for pre-pregnancy weight? Were those values self-reported?

Response to Reviewer 2

The BMI was calculated based on a height and weight measured by the midwife at the first antenatal clinic visit. This is considered more accurate than self-reported values. The pre-pregnancy weight was not obtained. This would not be feasible and previous studies have shown that self-reported weights are inaccurate [7] . A study conducted previously by our research group studied the weight, calculated BMI and body composition of 1000 women in the first trimester of pregnancy. The mean weight, BMI and body composition measures did not differ when the women were stratified according to number of weeks gestation from 5-13 weeks [8]. These references are included in the updated manuscript.

The text in the methods was updated as follows (page 6 line 7-18);

“The maternal weight and height were measured by a midwife at the first antenatal visit and the Body Mass Index (BMI) was calculated and categorized. This is more accurate than a self-reported pre-pregnancy weight. [7] Body weight, BMI and body composition have been shown to not vary significantly across the first trimester in a study of 1000 women. [8]

Page 5, line 10-12: Do not include birth outcomes in this analysis. They are not part of the primary research question (specifically, does dietary composition during pregnancy differ between smokers and non-smokers?). Also, by including those outcomes, you make the reader question why you didn't do a more extensive analysis looking at dietary quality and smoking status on birth outcomes.

Response to Reviewer 2

The infant birth weight is significantly different when comparing non-smokers and current smokers. We feel that this is an important secondary outcome of the research. This paper highlights the dietary inadequacies of smokers and strengthens the case for screening women in early pregnancy and offering smoking cessation support. An increased birth weight is an important clinical outcome associated with non-smoking status compared to infants of current smokers and further strengthens the argument for concentrating efforts in antenatal care to identify smokers and support smoking cessation.

There are many other birth outcomes included in the study (such as gestational age at delivery, preterm birth rate, rate of small for gestational age and caesarean section rate.) There were no significant differences in these when comparing smokers and non-smokers. They have been deleted from Tables 1 and 2 to simplify the presentation of the data.

Reviewer 2 Comment

Page 6, line 5: “Tests for statistical significance were performed at the 95% level (i.e., $p < 0.05$)”. That is incorrect – 0.05 is the level of statistical significance. 95% is the margin of error.

Response to Reviewer

This has been corrected in the text to (page 7, line 35-38)

“Tests for statistical significance were performed with the level for significance set at $p < 0.05$.”

Reviewer 2 comment

The Methods needs to include the details about comparing dietary intakes to WHO reference levels. Or, move that comparison out of the Results and into the Discussion.

Response to Reviewer

The European Food Safety Authority guidelines was our reference point for the nutrient requirements. The methods are updated to include mention of this on (page 6, line 33-40) and the reference for this updated accordingly.

“The nutrient requirement reference levels for pregnancy (either average requirement, population reference intake or adequate intake) were obtained from the European Food Safety Authority (EFSA) guideline [10]. Levels for Sodium, Chloride and Vitamin E are not established.”

Reviewer 2 comment

The Methods needs to include the details on how “plausible energy intake reporters” were defined (as reported in the Results).

Response to Reviewer

Thank you for this constructive comment. The methodology used to identify plausible energy intake reporters and exclude likely dietary under-reporters has been elaborated on in the methods on page 6, line 42-55. (The in-text references number 8 (Goldberg et al) and 9 (Black) refer to this methodology).

“To overcome the possible limitation associated with nutritional surveys of under-reporting, a calculation was applied to identify any likely under-reporters which is based on energy intake, basal metabolic rate and physical activity level. [11,12] Based on the reported physical activity level (PAL) the lowest plausible PAL threshold (from 1.45-2.20) was determined. If the energy intake (EI) to basal metabolic rate (BMR) ratio fell below the plausible threshold for their PAL then they were identified as a dietary under-reporter. Plausible energy intake reporters could be identified in this way.”

6. Results

Reviewer 2 comment

When discussing the sample – the sample is the study *sample*, not the study population. For example, page 6, line 18: “The characteristics of the study the study population is shown...”. That should be written as: “The characteristics of the study the study *sample* *are* shown...”.

Response to Reviewer 2

This has been updated in the text to state *sample* rather than population in the following sections

Strengths and limitations

Methods

Results

Discussion

Table 1 caption and in table.

Reviewer 2 comment

Line 38: “Table 3 shows theand analyses the...” I think that should read “analyzes”. However, the table doesn’t analyze the data. Thus, I’d rewrite as “Table 3 presents the dietary intakes overall and stratified by smoking status.”

Response to Reviewer

Thank you for this comment. This has been updated in the text as follows on page 8, line 26-27

“Table 3 presents the dietary intakes overall and stratified by smoking status”

Reviewer 2 comment

Page 7, lines 2-7: “ There was no relationship”. Drop the line “This may have been due to a small sample size.” That is not relevant here (move that to the Discussion if needed). However, why do you think diet would differ with smoking intensity? I would consider dropping that aspect of the analysis.

Response to Reviewer

Thank you for the comment. The line “This may have been due to a small sample size” has been omitted.

Reviewer 2 comment

Page 7: When discussing Supplementary Tables 1 and 2, the reference to combining never and former smokers is confusing. Do all of the following analyses combine those two groups? If they do – make sure each table has a footnote to clarify that, and move that detail to the Methods. In fact, you can state in the Methods that because preliminary analyses documented that dietary composition was similar between never smokers and former smokers (Supplemental table X), analyses combined never and former smokers into one “non-smokers” group.

Response to Reviewer

The methods have been updated to mention that never smokers and former smokers are combined into one group for the analyses on page 7, line 3-8 and the detail on combining them originally mentioned in the results section has been deleted.

“Preliminary analyses conducted demonstrated that dietary composition was similar between never smokers and former smokers (Supplementary tables 1 and 2), therefore, analyses combined never and former smokers into one “non-smokers” group.”

Footnotes have been added to Tables 2,3,4 and Supplementary tables 3 and 4 to clarify this.

Reviewer 2 comment

In all tables, include the tests used for statistical significance as a footnote (i.e., define where the P-values came from).

Response to Reviewer

The footnotes of all tables have been updated to include the significance level (0.05) and the statistical tests used as applicable – chi-square test for categorical variables, independent samples t-test for normally distributed continuous variables and Kruskal-Wallis non-parametric test for non-normally distributed continuous variables.

7. Discussion

Reviewer 2 comment

Page 8, lines 26-44. These analyses looked at what women ate. It does not look at “availability” of nutrients (which implies bioavailability). Thus, quitting smoking does not impact the availability of any nutrient. Likely, women quit smoking and made other positive health changes – including improving their diets.

Response to Reviewer

Thank you for this comment. The discussion has been updated on page 10, line 28-36 to omit the mention of availability and instead mention the likely improvement in maternal diet.

“The findings in former smokers also suggests that cessation of smoking during pregnancy results not only in the fetal avoidance of cigarette toxins but also in likely improvements in the maternal diet with increasing intakes of important micronutrients such as iron, zinc, riboflavin, niacin and vitamin B12 ($p<0.05$ for all).”

Reviewer 2 comment

Move the discussion of strengths and limitations to the penultimate paragraph.

Response to Reviewer

This has been updated with strengths and limitations moved to the penultimate paragraph of the discussion.

Reviewer 2 comment

This discussion includes an excessive discussion of past studies which have reported differences in dietary intakes among smokers and non-smokers. This information (across multiple paragraphs) can be succinctly summarized with a brief statement such as “Our findings that smokers have a poor dietary composition relative to non-smokers is consistent with many previous studies, including those among pregnant women (references) and the general adult population (references).”

Response to Reviewer

The discussion of studies comparing dietary intake of non-smokers and smokers has been shortened significantly with the previous detailed discussion of the references in the original version of the manuscript omitted and instead the following statement included (with reference to these studies as numbers 15-20 respectively) (page 10, line 37-42)

“Our findings that smokers have a poor dietary composition relative to non-smokers is consistent with many previous studies, including those among the general adult population [15-18] and pregnant women [19,20].”

Reviewer 2 comment

The Discussion needs to include some discussion of how smoking is correlated with other poor health behaviors, including poor dietary habits. Specifically, please recognize that it is not quitting smoking that changes a woman’s dietary intake, but other lifestyle changes that may covary with quitting smoking. You could discuss the importance of quality nutrition during pregnancy, and the implications on maternal and infant health. Then comment on the dangers of smoking during pregnancy – and then hypothesize about the increased risks women may face if they smoke and also have a poor diet quality.

Response to Reviewer

The discussion includes the NHANES study regarding dietary habits of smokers and has been updated to include a further reference regarding the poor health behaviours associated it smoking.

This is on page 11 line 7-23

A Swiss population study of 2301 men and 2306 women reported that current heavy smokers consumed less total vegetable proteins, carbohydrates, fibre, betacarotene, fruit and vegetables compared to non-smokers (p values all significant). (Morabia et al). They also drank more alcohol and coffee ($p < 0.0001$ and $p < 0.005$ respectively). The female smokers also consumed less complex carbohydrates ($p < 0.002$) and less iron ($p < 0.02$). As in this study, the diet of ex-smokers was similar to never smokers. This demonstrates that it is likely that quitting smoking is associated with other positive lifestyle modifications including improvement of diet.

The next part of the discussion details the importance of dietary quality in pregnancy referencing with a risk reduction preterm birth [26] , improved growth and vitamin C and RBC folate levels (5) and the

reference to Barker's hypothesis with undernutrition in the fetal period linked to an increase risk of type two diabetes mellitus, hypertension and coronary heart disease in later life (4,27)

This is followed by discussion of the increased risk to women who continue to smoke in pregnancy including an increased risk of neural tube defects [29] and a further reference incorporated on page x line y, to show the increased risk of preterm delivery, intrauterine fetal demise, lower birth weight and neonatal death and a dose response relationship was detected. [28] (page 12 lines 52-55 and page 13 lines 3-10)

"A Canadian study of the period 2001-7 of 1,646 self-reported smokers compared to 19,292 non-smokers reported an increase in adverse perinatal and neonatal outcomes. There was an increased risk of preterm delivery (22.2% vs 12.4%, $p < 0.05$), intrauterine fetal demise (OR 2.4 (95% CI 1.4-4.2), a mean birth weight approximately 200 grams lower in the current smokers ($p < 0.05$) and an increased risk of neonatal death (1.2% vs 0.6, $p < 0.05$). [28]"

Reviewer 2 comment

It is also possible that women who smoke during pregnancy may benefit the most from a dietary intervention (see Emond et al, Chatzi et al), and you noted in the Conclusions.

Response to Reviewer 2

The point alluded to in the conclusion that dietary intervention may benefit women who continue to smoke in pregnancy is elaborated in the discussion incorporating the references for same. (See page 13 line 40-54)

Women who continue to smoke in pregnancy may benefit from dietary intervention. A study of 862 women and infant pairs assessed maternal dietary quality at 24-28 weeks gestation. Lower dietary quality was associated with lower educational attainment, maternal smoking, obesity (pre-pregnancy) and a lack of exercise during pregnancy. It was found that increased dietary quality was associated with a reduced likelihood of a small for gestational age infant. Furthermore, a higher diet quality was positively associated with birth weight amongst former and current smokers. [31]

and page 14, line 3-14)

A study of two mother and infant cohorts (2461 pairs in Spain and 889 pairs in Greece) found that a high Mediterranean diet (MD - vegetables, legumes, fruit, nuts, cereals, seafood and fish and dairy products) adherence was associated with a lower risk of a growth restricted infant (risk ratio 0.5, 95% CI 0.3-0.9). When the analysis was stratified according to smoking status, higher MD adherence was associated with higher birth weight and birth length in the infants of mothers who continued to smoke. [32]

Reviewer 2 comment

Include some mention as to whether you think dietary intake (as measured at the 1st trimester) is likely to be consistent throughout pregnancy.

The Conclusions paragraph is well written.

Response to Reviewer

We agree that the first trimester diet may be influenced by nausea and vomiting compared to the second and third trimester. There are limited studies in the literature that have assessed the dietary quality of the same cohort of pregnant women in all three trimesters. The discussion has been updated to reference a study that demonstrated the dietary intake in the first trimester did not change significantly in the second and third trimester. (page 14, line 38-49).

“Dietary information was gathered in the first trimester only and it is not known if dietary choices changed as pregnancy advanced. An Irish study of 285 women who completed a three-day food diary in each trimester used cluster analysis to identify two major dietary patterns, “Unhealthy” and “Health conscious”. Comparing the energy and nutrient intakes from the first trimester to the third trimester they found no significant change in either cluster. [33]”

VERSION 2 – REVIEW

REVIEWER	Monica Tarcea University of Medicine and Pharmacy Tirgu Mures, Romania
REVIEW RETURNED	06-Mar-2018
GENERAL COMMENTS	Congratulations for your study, its interesting, well written and detailed. I recommend to delete Table 1, we have to focus on those 398 women, not 502. If its possible, avoid References older than 2000 and with only

	abstract to read about, you need up-to-date articles.
REVIEWER	Jennifer A. Emond Dartmouth College
REVIEW RETURNED	26-Mar-2018

GENERAL COMMENTS	The manuscript was revised and it is improved. I have minor suggestions to improve the presentation of the findings. Table 1 currently presents key sample characteristics between the entire sample (n=502) and the analytical sample with dietary data (n=398). There are only two statistically significant differences. Table 1 is not critical to the paper, it slows the reader down, and thus Table 1 is not an efficient use of a table. Instead, a brief summary of the differences between the enrolled sample and the analytical sample can be summarized in text: “When compared to the entire of 502 women enrolled in the study, women with dietary data were less likely to be nulliparous (P=0.01) or Irish (P<0.001). Furthermore, if the authors feel that these differences are important, plus discuss the limitations related to these differences in the Discussion. Discussion Page 11, lines 27-29: “...but our observations raise the possibility that dietary deficiencies may contribute, for example, to the risk of fetal growth restriction.” This is a bold statement that is not supported with the data, nor is it supported here in the Discussion with past studies. If you do report differences in birth weight by smoking status in the results and make a statement such as this in the discussion—test that hypothesis. For example, include a mediation analysis to test that hypothesis. Else, move this statement later into the discussion where potential implications are discussed. For example, as the lead line to page 15, line 45, “Women who continue to smoke...” Then, page 11 lines 26-42 can be dropped. This background doesn't add to this paper. In fact, lines 29-35: “The decrease in total dietary folate in smokers was also associated with a decreased rate of initiation of pre-pregnancy Folic Acid supplementation which increases the risk of a pregnancy complicated by a Neural Tube Defect (NTDs).” Folic acid supplementation is not related to dietary patterns and thus diet quality...smokers could just take the supplement and reduce risk of NTD. Importantly, this statement regarding supplementation and NTD does not follow from “...dietary deficiencies may contribute, for example, to the risk of fetal growth restriction.” Page 10 lines 35-42: “The findings in former smokers also suggests that cessation of smoking during pregnancy results not only in the fetal avoidance of cigarette toxins but also in likely improvements in the maternal diet with increasing intakes of important micronutrients such as iron, zinc, riboflavin, niacin and vitamin B12 (p<0.05 for all).” This language still implies causality that quitting smoking improves dietary intake: “...but also in likely improvements in the maternal diet...”. That is not true. For example, you did have dietary intake pre-pregnancy. What if those who were able to quit early in pregnancy (i.e., they quit before the first visit) were lighter smokers and had better diets than heavier smokers who did not quit? As
---

	noted above, because you are summarizing your key findings here, I would drop this text. You do bring this topic up later, which is fine.
--	---

VERSION 2 – AUTHOR RESPONSE

Reviewer(s)' Comments to Author:

Reviewer: 1

Reviewer Name: Monica Tarcea

Institution and Country: University of Medicine and Pharmacy Tirgu Mures, Romania

Please state any competing interests or state 'None declared': None to declare.

Please leave your comments for the authors below

Congratulations for your study, its interesting, well written and detailed.

Reviewer 1 comment

I recommend to delete Table 1, we have to focus on those 398 women, not 502.

Response

This comment was also made by reviewer 2 and this table has been removed and the text already outlines the significant results that were in this table.

Page 8 lines 18-21

“...those who did fill out the dietary questionnaire were more likely to be nulliparous ($p=0.012$) and Irish-born ($p<0.001$).”

Tables 2, 3 and 4 in the previous version are now Tables 1, 2 and 3 and all instances in the text have been updated accordingly.

Reviewer 1 comment

If it's possible, avoid References older than 2000 and with only abstract to read about, you need up-to-date articles.

Response

The following references before the year 2000 have been changed to more up to date articles

4. Barker DJP, Eriksson JG, Forsen T, Osmond C. Fetal origins of adult disease: strength of effects and biological basis. *Int J Epidemiol* 2002;31:1235-1239

15. Lloveras G, Ribas LB, Ramon JM, Serra LM, Román BV. Food consumption and nutrient intake in relation to smoking. *Medicina clinica*. 2001;116(4):129-32.

18. Haibach JP, Homish GG, Giovino GA. A longitudinal evaluation of fruit and vegetable consumption and cigarette smoking. *Nicotine & Tobacco Research*. 2012 May 21;15(2):355-63.

19. Cogswell ME, Weisberg P, Spong C. Cigarette smoking, alcohol use and adverse pregnancy outcomes: implications for micronutrient supplementation. *J Nutr*. 2003;133(5):1722S-1731S.

30. Crider KS, Qi YP, Devin O, Tinker SC, Berry RJ. Modeling the impact of folic acid fortification and supplementation on red blood cell folate concentrations and predicted neural tube defect risk in the United States: have we reached optimal prevention? *Am J Clin Nutr*. 2018
<https://doi.org/10.1093/ajcn/nqy065>

In two instances, the original reference has been retained;

For reference 11 – Goldberg at al, 1991 - a more up to date reference is not available as this is the original paper for the derivation of the formula to identify energy under-reporters.

For reference 21, this study reports on the second US National Health and Nutrition Examination Survey. This survey reported on the dietary deficiencies of smokers compared to non-smokers and studied over 11,000 participants. The subsequent NHANES III reported lower vitamin C and vitamin E levels and antioxidants but did not report on detailed dietary intake as in the NHANES II so this reference has been retained for this reason.

Reviewer: 2

Reviewer Name: Jennifer A. Emond

Institution and Country: Dartmouth College

Please state any competing interests or state 'None declared': None declared.

Please leave your comments for the authors below

The manuscript was revised and it is improved. I have minor suggestions to improve the presentation of the findings.

Reviewer 2 comment

Table 1 currently presents key sample characteristics between the entire sample (n=502) and the analytical sample with dietary data (n=398). There are only two statistically significant differences. Table 1 is not critical to the paper, it slows the reader down, and thus Table 1 is not an efficient use of a table. Instead, a brief summary of the differences between the enrolled sample and the analytical sample can be summarized in text: "When compared to the entire of 502 women enrolled in the study, women with dietary data were less likely to be nulliparous (P=0.01) or Irish (P<0.001). Furthermore, if the authors feel that these differences are important, plus discuss the limitations related to these differences in the Discussion.

Response

This comment was also made by reviewer 1 and this table has been removed and the text already outlines the significant results that were in this table.

Page 8 lines 18-21

"...those who did fill out the dietary questionnaire were more likely to be nulliparous (p=0.012) and Irish-born (p<0.001)."

Tables 2, 3 and 4 in the previous version are now Tables 1, 2 and 3 and all instances in the text have been updated accordingly.

Reviewer 2 comment

Discussion

Page 11, lines 27-29: "...but our observations raise the possibility that dietary deficiencies may contribute, for example, to the risk of fetal growth restriction."

This is a bold statement that is not supported with the data, nor is it supported here in the Discussion with past studies. If you do report differences in birth weight by smoking status in the results and make a statement such as this in the discussion—test that hypothesis. For example, include a mediation analysis to test that hypothesis. Else, move this statement later into the discussion where potential implications are discussed. For example, as the lead line to page 15, line 45, "Women who continue to smoke..."

Response

This statement "...but our observations raise the possibility that dietary deficiencies may contribute, for example, to the risk of fetal growth restriction." has been removed.

This line now reads

PAGE 10 LINES 18-23

"Maternal smoking is strongly associated with adverse clinical outcomes for both the woman and her offspring and this study highlights that dietary quality and deficiencies are another area of concern."

Reviewer 2 comment

Then, page 11 lines 26-42 can be dropped. This background doesn't add to this paper. In fact, lines 29-35: "The decrease in total dietary folate in smokers was also associated with a decreased rate of initiation of pre-pregnancy Folic Acid supplementation which increases the risk of a pregnancy complicated by a Neural Tube Defect (NTDs)." Folic acid supplementation is not related to dietary patterns and thus diet quality...smokers could just take the supplement and reduce risk of NTD. Importantly, this statement regarding supplementation and NTD does not follow from "...dietary

deficiencies may contribute, for example, to the risk of fetal growth restriction.”

Response

Thank you for the constructive feedback. This section has been removed from the text.

Reviewer 2 comment

Page 10 lines 35-42: “The findings in former smokers also suggests that cessation of smoking during pregnancy results not only in the fetal avoidance of cigarette toxins but also in likely improvements in the maternal diet with increasing intakes of important micronutrients such as iron, zinc, riboflavin, niacin and vitamin B12 (p<0.05 for all).”

This language still implies causality that quitting smoking improves dietary intake: “...but also in likely improvements in the maternal diet...”. That is not true. For example, you did have dietary intake pre-pregnancy. What if those who were able to quit early in pregnancy (i.e., they quit before the first visit) were lighter smokers and had better diets than heavier smokers who did not quit? As noted above, because you are summarizing your key findings here, I would drop this text. You do bring this topic up later, which is fine.

Response

This section has been removed from the text.

VERSION 3 – REVIEW

REVIEWER	Monica Tarcea University of Medicine and Pharmacy Targu Mures
REVIEW RETURNED	21-May-2018

GENERAL COMMENTS	No comments
-------------

REVIEWER	Jennifer A. Emond Dartmouth College, USA
REVIEW RETURNED	29-May-2018

GENERAL COMMENTS	The manuscript is quite strong and will be of interest to many clinicians and public health researchers. I thank the authors for acknowledging our work in this field. However, for reference 31, the name is "Emond", not "Edmond."
--